# Regulation, Activation and Function of Caspase-11 during Health and Disease

**DOI:** 10.3390/ijms22041506

**Published:** 2021-02-03

**Authors:** Aidan Agnew, Ciara Nulty, Emma M. Creagh

**Affiliations:** School of Biochemistry and Immunology, Trinity Biomedical Sciences Institute, Trinity College Dublin, D02R590 Dublin, Ireland; agnewa@tcd.ie (A.A.); nultyci@tcd.ie (C.N.)

**Keywords:** Caspase-11, non-canonical inflammasome, pyroptosis, Gasdermin D, Gram-negative bacterial infection, sepsis

## Abstract

Caspase-11 is a pro-inflammatory enzyme that is stringently regulated during its expression and activation. As caspase-11 is not constitutively expressed in cells, it requires a priming step for its upregulation, which occurs following the stimulation of pathogen and cytokine receptors. Once expressed, caspase-11 activation is triggered by its interaction with lipopolysaccharide (LPS) from Gram-negative bacteria. Being an initiator caspase, activated caspase-11 functions primarily through its cleavage of key substrates. Gasdermin D (GSDMD) is the primary substrate of caspase-11, and the GSDMD cleavage fragment generated is responsible for the inflammatory form of cell death, pyroptosis, via its formation of pores in the plasma membrane. Thus, caspase-11 functions as an intracellular sensor for LPS and an immune effector. This review provides an overview of caspase-11—describing its structure and the transcriptional mechanisms that govern its expression, in addition to its activation, which is reported to be regulated by factors such as guanylate-binding proteins (GBPs), high mobility group box 1 (HMGB1) protein, and oxidized phospholipids. We also discuss the functional outcomes of caspase-11 activation, which include the non-canonical inflammasome, modulation of actin dynamics, and the initiation of blood coagulation, highlighting the importance of inflammatory caspase-11 during infection and disease.

## 1. Introduction

Caspases are a family of cysteine aspartic proteases that have been classically sub-divided into those involved in apoptosis or inflammation. However, research over the last two decades has revealed that this family of proteases is central to coordinating and integrating signals that result in cytokine maturation, inflammation, and cell death processes including apoptosis, pyroptosis, and necroptosis [1]. Caspases are expressed in most cells as inactive monomeric zymogens (pro-caspases), requiring processing and heterodimerization to facilitate their activation. The enzymatic activity of caspases is ruled by a dominant specificity for aspartic acid containing proteins and a cysteine side chain that acts as a catalytic nucleophile to employ peptide cleavage [2]. The inflammatory caspases consist of four human members, caspases-1, -4, -5, and -12. Inflammatory caspase zymogens share a conserved N-terminal pro-domain and a C-terminal protease domain. The long pro-domains of inflammatory caspases consist of CARD protein interaction domains, while the protease domain consists of a large (20 kDa) and small (10 kDa) subunit that contains the catalytic cysteine residue. Caspase-1 is the prototypical inflammatory caspase, first identified as interleukin-1β converting enzyme (ICE) [3,4]. Caspase-4 and -5 are very similar in sequence, having arisen due to a gene duplication event. Caspases-4 and -5 are the human orthologues of murine caspase-11, sharing with it 68% and 47% amino acid sequence identity, respectively. Caspase-12 is predominantly expressed as a truncated version in humans, containing only the N-terminal CARD domain. Full-length caspase-12 is expressed in approximately 20% of people of African descent, where it has been linked to increased severity of sepsis. It is therefore hypothesized that evolutionary loss of the caspase-12 catalytic domains confers a selective advantage by increasing resistance to sepsis [5]. The human inflammatory caspase genes are located on chromosome 11q22.2-q22.3 [6], while the murine genes are located on syntenic regions of mouse chromosome 9A1 [7].

Caspase-1 was first identified as a pre-aspartate-specific protease involved in the cleavage of the pro-IL-1β (31 kDa) precursor to its biologically active form, IL-1β (17 kDa) [4]. IL-1β and IL-18 are potent proinflammatory cytokines that induce fever and IFNγ secretion, respectively. Their production is tightly regulated, with both cytokines being synthesized as immature precursors that require caspase-1 mediated processing for their maturation. Caspase-1 activation occurs through inflammasomes—multi-protein complexes that form in response to pathogenic microbes or sterile cell damage, following cellular detection of specific pathogen-associated molecular patterns (PAMPs) or danger-associated molecular patterns (DAMPs). In addition to the proteolytic activation of IL-1β and IL-18, inflammasome activation also results in pyroptosis, an inflammatory form of cell death. The best characterized inflammasome to date is the canonical NLRP3 inflammasome, which requires an initial PAMP-mediated priming step, prior to its oligomerization and activation, which is triggered by a diverse range of cytosolic PAMPs/DAMPs [8]. Activated caspase-11 (and human caspases-4/-5) are capable of activating the NLRP3 inflammasome through an alternative mechanism (described below), termed the non-canonical inflammasome. Although caspase-1 is the prototypical member of the inflammatory caspase family, caspase-11 is emerging as an important inflammatory regulator, controlling the activation of the non-canonical inflammasome in addition to other effector functions, which will be discussed in this review.

## 2. Caspase-11 Structure

Faucheu and colleagues first identified caspase-4 and caspase-5 as members of the ICE (caspase-1) family in 1996 [9]. The homology of caspases-4 and -5 to caspase-1 included their conserved catalytic pentapeptide (Gln-Ala-Cys-Arg-Asp) site, which harbors the catalytic cysteine residue; aspartate-binding pockets; and the fact that they are synthesized as inactive precursor zymogens with a C-terminal CARD domain [10]. Caspase-4 demonstrated auto-proteolysis and cleavage of p30 caspase-1 precursors [9]. Caspase-5 also demonstrated protease activity of its own precursor, dependent on the active site cysteine (residue 245) [11]. Once proteolytically processed, both caspase-4 and -5 form heterodimeric active enzymes, incapable of pro-IL-1β processing but capable of inducing cell death upon overexpression [9,10,11]. Caspase-11 auto-proteolysis is essential for its activation and the subsequent regulation of downstream events. Caspase-11 activation, induced experimentally via its overexpression or cytoplasmic LPS stimulation, induces two auto-proteolysis events, resulting in the removal of the N-terminal CARD domain from the large subunit and separating the large and small catalytic domains (Figure 1) [12]. Elegant studies from the Kayagaki group have demonstrated that caspase-11 processing at aspartic acid 285, which separates the large and small subunits, represents a non-redundant auto-processing site, essential for the activation of downstream events—pyroptotic cell death and secretion of IL-1β and IL-18 [13]. Conserved residues in human caspase-4 (Asp 289) and other species suggest that this mechanism for caspase-4/11 auto-proteolysis is evolutionarily conserved [13].

Similar to the canonical inflammasome, the caspase-11 activated non-canonical inflammasome leads to pyroptosis and ultimately release of mature proinflammatory IL-1β and IL-18, albeit by a different mechanism [14]. Caspase-11/-4/-5 mediated activation of the non-canonical inflammasome is triggered by its ability to directly stimulate pyroptosis, via its direct cleavage of Gasdermin D (GSDMD), to generate an N-terminal pore-forming fragment (GSDMD-N), as detailed later in this review. However, prior to its activation, caspase-11 requires a transcription-dependent “priming” signal to upregulate its cellular expression. Although caspase-11 is the murine orthologue of human caspases-4 and -5, there are subtleties in their expression and regulation. While the expression of caspase-5 and -11 require a priming signal, caspase-4 is constitutively expressed at basal levels within tissues, which may further increase following stimulation [15,16].

## 3. Caspase-11 Expression

Caspase-11 is regulated at two distinct levels; the level of inactive precursor procaspase-11 expression (short-hand referred to as caspase-11) and the level of activation, which involves proximity-induced dimerization of caspase-11 monomers. The autoproteolysis and heterodimerization of caspase-11 is mediated by the direct interaction of its CARD domain with LPS, as will be described in detail below [17,18]. Firstly, this review will discuss the factors that regulate the expression of caspase-11 (Table 1), followed by an account of the variables that affect caspase-11 activation.

The principal stimuli responsible for regulating the expression levels of inflammasome machinery, including caspase-11, are PAMP levels and cytokine levels. The extracellular detection of PAMPs by transmembrane pattern recognition receptors (PRRs) constitutes the first step of immune cell priming, which induces the upregulation of immune effector proteins. Toll-like receptors (TLRs) are the major class of PRR responsible for the priming of inflammasome machinery and inflammatory caspases. Upon PAMP ligand binding to the extracellular leucine-rich repeat structure of plasma membrane-associated TLRs, the cytoplasmic toll-IL-1 receptor domain (TIR domain) of the TLR molecule associates with the TIR domain of an adaptor protein, called MYD88. MYD88-dependant TLR signaling results in the downstream activation of NF-κB (Figure 2) [33]. Schauvliege et al. identified an NF-κB binding site in the promoter sequence of caspase-11, and through the use of a library of luciferase reporter gene vectors with mutations inserted into different positions of the promoter sequence, found that LPS-induced caspase-11 expression was partially mediated by NF-κB activity [31]. Various different PAMPs can therefore upregulate caspase-11 through the signaling cascades initiated by different TLRs (flagellin through TLR5, viral dsRNA through TLR3, bacterial CpG-rich DNA through TLR9), which converge on the action of NF-κB [19].

LPS-mediated upregulation of caspase-11 in bone marrow-derived macrophages (BMDMs) was first demonstrated by the Yuan group in 1998 [34]. TLR4 binds extracellular LPS and signals through the adaptors Mal and MYD88 to result in NF-κB activation [33,35], but following its endosomal uptake into the cell, TLR4 also signals through the TRIF adaptor protein [21]. Sander et al. demonstrated, using gene microarray analysis, that *wild type* (WT) and *Trif*^−/−^ BMDM display significantly different gene expression profiles in response to LPS, making it clear that a significant portion of TLR4-inducible genes were expressed independently of MYD88. This study identified several MYD88-dependent genes such as IL-6, NLRP3, and IL-1β, whose gene expression did not require TRIF, but also showed that interferon-β (IFN-β) expression was TRIF-dependent and did not require MYD88. The study also showed that pyroptosis, following infection with viable *E. coli*, was completely blocked in *Trif*^−/−^ BMDM [22]. This study provided evidence to suggest that expression of the endogenous LPS sensor, caspase-11, is TRIF-dependent. Rathinam et al. confirmed this observation by demonstrating that caspase-11 expression was significantly abrogated in *Trif*^−/−^ BMDM infected with *E. coli*, and was also significantly impaired in *Ifnar*^−/−^ BMDMs [20]. TRIF-signaling downstream of LPS-bound TLR4 leads to the activation of the IFN-regulatory factors IRF3/7 [29], which are primarily responsible for the upregulation of type I IFNs (IFNα/β) (Figure 2). Type I IFNs then stimulate IFNAR in a paracrine or autocrine manner, resulting in the formation of the ISGF3 transcriptional complex, consisting of STAT1, STAT2, and IRF9 [30]. This complex is responsible for the transcriptional upregulation of caspase-11 and other genes containing gamma-activated promoter sequences (GAS) in response to LPS or IFNα/β. This TLR4-TRIF-IRF3/7-IFNβ pathway demonstrates that LPS and type I IFNs act in conjunction with one another to regulate the expression of caspase-11. Studies carried out in our lab have shown that IL-1β is also capable of inducing caspase-11 upregulation in murine BMDMs. Although IL-1β signals through the interleukin 1 receptor (IL-1R), resulting in the activation of NF-κB [36], we have shown that IL-1β-mediated upregulation of caspase-11 is impaired in *Ifnar*^−/−^ BMDM, providing evidence that type I IFNs govern caspase-11 expression downstream of IL-1β [23].

Another cytokine that acts as a major regulator of caspase-11 expression is IFN-γ. Research from Kate Fitzgerald’s group has shown that impaired caspase-11 expression in *Ifnar*^−/−^ BMDM could be rescued to levels that were comparable to that of WT cells, by stimulation with IFN-γ as opposed to infection with Gram-negative *E. coli* [20]. IFN-γ binds to the IFN-γ receptor complex (IFNGR) as a homodimer, and subsequent JAK-STAT signaling results in the formation of a phospho-STAT1 homodimer, which translocates to the nucleus and partakes in ISG upregulation (Figure 2) [32]. Other clear evidence that points towards IFN-γ being a principal regulator of caspase-11 expression is the presence of a STAT1 binding site in the promoter sequence of caspase-11 [31].

Although type I IFN (via direct stimulation or indirectly via LPS or IL-1b) and IFN-γ are the major regulatory stimuli responsible for the upregulation of caspase-11 experimentally in BMDM, recent studies have identified the presence of additional stimulatory pathways that can partially modulate caspase-11 expression in the context of certain diseases. One study used both murine BMDM and the murine macrophage cell line RAW267.4 (RAW) to demonstrate that the alternative complement pathway significantly controls caspase-11 expression. The Monack group used a CRISPR-Cas9 genome editing system to generate a knockout library of guide RNA sequences (gRNA) against 19,150 protein-encoding genes in RAW cells, in order to identify mediators of caspase-11-dependant cell death. They identified that the gene encoding for carboxypeptidase B1 (Cpb1) mediated the expression of caspase-11 in response to LPS or IFNβ, but not IFNγ [26]. Cpb1 is localized to the extracellular domain of the C3a receptor (C3aR) and is responsible for the modification of the complement protein cleavage product C3a, to C3a-desArg, which acts as a ligand for C3aR [27,28]. The C3aR signaling pathway results in the downstream activation of MAPK kinase p38 [37], resulting in the non-canonical transcription of pro-inflammatory genes, including caspase-11. In the context of Gram-negative bacterial infection and experimental sepsis, MAPK-P38 activity was shown to be enhanced downstream of both IFNAR and TLR signaling, providing a mechanism for the effect of Cpb1 on caspase-11 transcription (Figure 2) [26]. Thus, the Cpb1-C3-C3aR signaling axis of the complement system represents an important amplification mechanism for pro-inflammatory signaling and caspase-11-mediated cell death during infectious disease.

In contrast to the role of complement in amplifying caspase-11 expression during murine sepsis models [26], prostaglandin E_2_ (PGE_2_) is capable of potently inhibiting LPS-induced IFNβ production in murine BMDMs, which has the subsequent effect of significantly impairing LPS-mediated caspase-11 expression [38]. PGE_2_ has been extensively studied for its protective effect in asthma [39], and the study by Zaslona et al. links these protective effects to the ability of PGE_2_ to negatively regulate caspase-4 expression in alveolar macrophages from asthma patients [38].

It is evident that the modulation of IFNβ expression has a huge effect on the expression of caspase-11, and one other specific example of an immune signaling pathway that controls caspase-11 expression through the regulation of IFNβ levels is the cyclic GMP-AMP synthase/stimulator of IFN genes (cGAS-STING) cytosolic DNA sensing pathway [24]. A recent study has shown that the cGAS-STING pathway mediates caspase-4 upregulation in retinal pigmented epithelium (RPE) cells of the eye during age-related macular degeneration [25]. Caspase-11 has also been shown to be upregulated via the same pathway in relevant murine disease models. Activation of the cGAS-STING pathway occurs in response to cytosolic accumulation of mtDNA following mitochondrial damage [25]. cGAS-STING activation results in IRF3-mediated IFNβ upregulation [24], which is then responsible for the upregulation and activation of caspase-4/11, causing the detrimental exacerbation of macular degeneration pathology through pyroptosis in the RPE [25].

## 4. Caspase-11 Activation

Caspase-11 is essentially an endogenous receptor for LPS, which binds to the CARD domain of caspase-11 through its lipid A tail moiety to regulate caspase-11 activation [17]. It was initially hypothesized that caspase-11 activation somewhat resembled caspase-1 activation through the canonical inflammasome. During canonical NLRP3 inflammasome assembly, the PYRIN-CARD-containing adaptor protein, ASC, binds procaspase-1 via CARD–CARD interactions, while simultaneously binding NLRP3 through pyrin domain interactions (PYD–PYD) [40]. Shi et al. endeavored to find a mammalian CARD domain-containing protein that could induce caspase-11 dependent cell death when transfected into the 293T cell line; however, they were unable to do so. During the process of preparing recombinant caspase-4/11 from *E. coli* cells, it was observed that the recombinant proteins were eluting from a gel filtration column as oligomers, suggesting that a bacterial component was inducing oligomerization of *E. coli* produced caspase-4/11. A series of pulldown assays using Flag-tagged caspase-11 and biotinylated immunostimulatory peptides such as LPS, MDP, and Pam3CSK4 identified LPS as a mediator of caspase-11 oligomerization. Further surface plasmon resonance (SPR) experiments identified very high binding affinity between the caspase-11 lipid A tail moiety and the CARD domain. Caspase-11 mutants that were incapable of binding to the LPS lipid A tail (with mutations generated at different positions in the CARD domain) also failed to mediate LPS-induced cell death in 293T cells [17]. Therefore, unlike caspase-1, caspase-11 activation does not require an upstream sensory complex and is directly activated by LPS. This seminal discovery has transformed the outlook on caspase-11 from being an inflammatory initiator caspase, to more of a hybrid between an immune effector and an intracellular PRR, with LPS being the principal agonist and regulator.

Thus, under conditions of Gram-negative bacterial infection, the hexa-acylated lipid A moiety of LPS binds directly to the caspase-11 CARD domain, leading to the oligomerization and proximity-induced activation of caspase-11, which is likely to be facilitated by the tendency of LPS to aggregate. Cytoplasmic penta-acylated and hexa-acylated LPS are capable of inducing caspase-11 activation; however, this is not observed for tetra-acylated LPS [41]. Under-acylated lipid A expressing pathogens, such as *Francisella novicida*, escape caspase-11 recognition in mice. However, this immune escape is not observed in humans. *F. novicida* triggers caspase-4, driving GSDMD-dependent pyroptosis and non-canonical inflammasome activation in human macrophages [42]. This study suggests that the human system has a broader reactivity than that of the mouse. The pathogen recognition differences between caspase-4 and caspase-11 stress the importance of disease models for predicting the clinical implications of non-conserved amino acid sequences.

## 5. Regulation of Caspase-11 Activity

A number of additional factors have been proposed to regulate the activity of caspase-11; these are discussed in detail below and are summarized in Table 2: 

GBPs and IRGMs.

LPS-induced caspase-11 activation has recently been found to be reliant on additional endogenous mediators. The phagocytosis of Gram-negative bacteria by innate immune cells such as macrophages results in caspase-11 activation, but additional studies demonstrated that caspase-11 dependent cell death was significantly impaired in BMDM that were deficient for a group of IFN-inducible GTPases, guanylate-binding proteins found on chromosome 3 (GBP^chro3^), which encode for GBP1, GBP2, GBP3, GBP5 and GBP7, thereby implicating these GTPases in the controlled lysis of vacuoles containing Gram-negative bacteria [43,44]. This in turn makes caspase-11 activation in response to Gram-negative bacterial infections dependent on GBP^chro3^.

Another group of GTPases that have been implicated in regulating the accessibility of LPS to caspase-11 are the immunity-related GTPases M clade (IRGM), which are dynamin-related membrane-remodeling proteins. One very recent study by Finethy et al. showed that a dysregulated expression profile of the IRGM isoforms *Irgm1*, *Irgm2,* and *Irgm3* could result in either dysfunctional LPS detection by caspase-11, or a susceptibility to endotoxemia in vivo [45]. IRGM2 was shown to significantly suppress LPS-induced inflammasome activation by downregulating the rate at which LPS became free in the cytosol from within phagocytotic vacuoles or outer-membrane vesicles (OMVs), and *Irgm2*^−/−^ mice were more susceptible to sepsis due to increased caspase-11 mediated inflammation. Although this study did not elucidate the exact mechanism by which IRGM2 regulates the inflammasome, it showed that the mechanism is caspase-11 dependent, that it is separate to caspase-11 upregulation (caspase-11 expression levels were comparable between IRGM-deficient and WT cells), and that transfection of BMDMs with LPS was capable of overcoming IRGM2 regulation [45]. As other studies have implicated IRGM protein involvement in the regulation of autophagosome-lysosome fusion [46], it is hypothesized that perhaps the different IRGM proteins are involved in regulating vacuole/phagosome stability, thereby regulating the accessibility of caspase-11 to endocytosed LPS [45].

HMGB1

Another protein that has been demonstrated as a key requisite for caspase-11 mediated pyroptosis and lethality in endotoxemia and bacterial sepsis is hepatocyte-released high mobility group box 1 (HMGB1) protein. HMGB1 is a multifunctional nuclear protein that acts as an alarmin or DAMP molecule when released from cells, but is known to play an important role in a wide range of cellular processes such as transcription, replication, DNA repair, and nucleosome formation [55]. HMGB1 has previously been shown to circulate in the bloodstream during endotoxemia and sepsis [47,48]. Studies have previously demonstrated conferred resistance to bacterial sepsis upon antibody-mediated inhibition of HMGB1 [47], and binding of LPS to HMGB1 [49]. These studies led Deng et al. to hypothesize that HMGB1 is delivering systemic LPS into the cytosol of macrophages and endothelial cells during sepsis and lethal endotoxemia [50]. The mechanism by which HMGB1 binds to extracellular LPS was shown to occur through via HMGB1 A-box and B-box motifs (both of which are needed to bind LPS) and internalized LPS through receptor for advanced glycation endproducts (RAGE)-mediated endocytosis. Using confocal microscopy and screening the cytosolic fractions of macrophages and endothelial cells for the presence of the lysosomal protease, cathepsin D, this study showed that HMGB1 induced lysosomal rupture following RAGE-mediated endocytosis, releasing LPS into the cytosol where it activated caspase-11 [50]. When looking at murine models of bacterial sepsis and lethal endotoxemia, HMGB1 must therefore be considered an important regulator of LPS-induced caspase-11 activation.

Oxidized Phospholipids

LPS is the only known agonist for caspase-11, binding the caspase-11 CARD domain through its lipid A tail moiety. Recent studies have identified the ability of oxidized phospholipids, a broad group of endogenous bioactive inflammatory modulators, to compete with intracellular LPS for caspase-11 binding, resulting in an antagonistic inhibition of the non-canonical inflammasome. One study demonstrated that the oxidized phospholipid 1-palmitoyl-2-arachidonoyl-*sn*-glycero-3-phosphorylcholine (oxPAPC) was capable of drastically inhibiting cytotoxicity and IL-1β release when co-transfected into BMDMs with low concentrations of LPS (2 μg mL^−1^ oxPAPC significantly abrogated LPS (100 ngml^−1^)-induced cytotoxicity, and 100 ng oxPAPC significantly impaired IL-1β release induced by 50 ng LPS). This effect was shown for BMDMs that were primed with LPS (TLR4 agonist), Pam3CSK4 (TLR2 agonist), and Poly(I:C) (TLR3 agonist). Using caspase-11 mutants (caspase-11^ΔCARD^ and caspase-11^C254A^) and pull-down assays, they also showed that oxPAPC competes with LPS for binding to the CARD and catalytic domains of caspase-11, unveiling a near-identical binding pattern for LPS and oxPAPC for caspase-11 [52]. Another study showed very similar results using another oxidized phospholipid, Stearoyl lysophosphatidylcholine (LPC), which blocked LPS binding to caspase-11 and significantly protected mice against lethal endotoxemia [53]. Despite the evidence that oxPAPC and related lipids inhibit caspase-11 activation, Zanoni et al. reported that very high levels of oxPAPC were capable of inducing oligomerization and activation of caspase-11 in murine bone marrow-derived dendritic cells (BMDC) [54]. This study used SPR analysis to show that oxPAPC binds to caspase-11 with reasonable affinity, albeit at a much lower binding affinity than that of LPS binding to caspase-11. However, there is still uncertainty over whether oxPAPC can be considered an activator of caspase-11, even at the very high concentrations demonstrated in BMDC. Chu et al. treated LPS-primed WT and *Casp-11*^−/−^ BMDC with oxPAPC and observed a slight but insignificant increase in IL-1β release in *Casp-11*^−/−^ BMDC, compared to WT, when treated with 200 μg mL^−1^ oxPAPC. The LPS/high dose oxPAPC-induced IL-1β release was not reduced in *Casp-11*^−/−^ BMDC, but was reduced in *Tlr4*^–/–^, *Asc*^–/–^, *Casp-1*^–/–^, and *Nlrp3*^–/–^ BMDC [52]. Any effects of oxidized phospholipids on caspase-11 therefore appear to be concentration, cell type, and context-dependent. When studies showed that PGE_2_ levels abrogated caspase-11 dependent pyroptosis in alveolar macrophages during asthma, the possibility that PGE_2_ or a similar arachidonic acid derivative was inhibiting caspase-11 through direct binding was investigated. However, this turned out not to be true, as PGE_2_ and other prostaglandins were found to be incapable of directly binding caspase-11 [38]. It therefore remains unclear whether endogenous lipid-based inflammatory mediators can be considered reliable activators of the non-canonical inflammasome.

cAMP and Protein Kinase A

One specific area of immunological research that receives substantial attention is the role of cellular metabolism during immunological activity. The interactions between metabolic and inflammatory pathways are important with regard to understanding how metabolites can be used to sensitize/desensitize our immune system during systemic inflammation. Previous studies have demonstrated that cAMP-PKA signaling can have a significant inhibitory impact on innate immune activation. PGE_2_ signaling through E prostanoid GPCR 4 (EP4) was shown to impinge on macrophage maturation, resulting in increased intracellular cAMP levels that activate PKA (cAMP has several effectors, but experiments using agonists for various cAMP effectors proved that PKA was necessary for the alleviation of macrophage maturation) [56]. Later studies expanded on this line of research by showing that PGE_2_-EP4 induced activation of PKA resulted in NLRP3 phosphorylation at specific serine residues, resulting in either rapid inhibition of NLRP3 activity or its subsequent ubiquitination [57,58]. There is now evidence to suggest that the cAMP-PKA signaling axis has a direct inhibitory effect on the non-canonical inflammasome as well as the NLRP3 inflammasome. Chen et al. showed that, just like PGE_2_-EP4 signaling and bile acid-TGR5 signaling, L-adrenaline can act on adrenoreceptor α 2B (ADRA2B), which is essentially coupled to the adenylyl cyclase enzyme ADCY4, resulting in elevated cAMP generation and activation of PKA. Immunoprecipitation experiments suggest that, upon L-adrenaline treatment of murine BMDMs, the PKA catalytic subunits (PRKACA and PRKACB) can form a complex with caspase-11, facilitating its inhibition via phosphorylation at serine sites. To further support the hypothesis that this signaling axis inhibited caspase-11 activation, the study also characterized the role of phosphodiesterase 8A (PDE8A), which degrades cAMP, and showed that it increased caspase-11 activation, pyroptosis, and IL-1β secretion in BMDMs, in response to cytosolic LPS [51].

## 6. Functional Effects of Caspase-11

The primary effector functions of caspase-11 are related to non-canonical inflammasome activity—resulting in alarmin release and pyroptosis, which subsequently triggers canonical inflammasome activation, IL-1β and IL-18 secretion in inflammatory and neighboring cells. Additional functions have also been attributed to caspase-11—including its ability to modulate actin dynamics, alter mitochondrial respiration, induce apoptosis, and trigger blood clotting in the context of bacterial infection or sepsis. The details of pyroptosis and some examples of alarmin release are described below. In addition, the merits of the additional functions proposed for caspase-11 are discussed in the following sections.

Pyroptosis

Gasdermin D (GSDMD) can be cleaved by either of the proinflammatory caspases, caspase-1 or caspase-11, to generate two fragments, the P20 and P30. The crystal structures of murine and human GSDMD were identified in 2019 [59], revealing that the C-terminal domain functions as an intrinsic inhibitor of the molecule. Following processing, the P30 fragment contains the functionally important GSDMD-N, which migrates to the plasma membrane where it oligomerizes with other GSDMD-N fragments to form membrane-permeable pores. These pores form the functional basis of pyroptotic cell death and non-specific cytokine release [60,61,62,63]. Caspase-1 has higher affinity for GSDMD processing than caspase-11 [64], meaning that once both caspases are active, caspase-1 would outpace caspase-11 at GSDMD cleavage. However, unlike caspase-1, which relies on its activation via inflammasome formation, caspase-11 functions to directly cleave GSDMD independently of inflammasome mediators, such as NLRP3, NLRC4, or ASC. The GSDMD pore has an inner and outer diameter of 18 and 28 nm, respectively [65]. Pore formation results in loss of ionic gradient, cell membrane rupture, and pyroptosis [66]. Caspase-11 induced GSDMD-N pore formation results in K^+^ efflux, which triggers inflammasome activation and caspase-1 processing, leading to proinflammatory cytokine maturation (Figure 3) [14]. In vitro, GSDMD-N colocalizes with membrane and mitochondrial lipid bilayers, with equal binding of GSDMD-N to the different liposome compositions [61]. However, GSDMD-N mediated liposome pore formation in the membranes of sub-cellular organelles has not yet been demonstrated in vivo. Research from the Lamkanfi group challenge the model that pre-assembled GSDMD-N pores insert into the plasma membrane to induce pyroptosis. Analyzing the kinetics of pyroptosis in single cells (LPS-transfected macrophages), they observed a conserved sequence of events, including Ca^2+^ influx and subcellular lysis, which preceded rupture of the plasma membrane [67]. They propose that GSDMD-N monomers insert into membranes individually, or as small oligomers that can subsequently assemble into higher order oligomers. This ultimately leads to late-stage events of pyroptosis—nuclear rounding and condensation, and the loss of plasma membrane integrity. Following GSDMD pore formation, mature IL-1β and organelle proteins, which serve as DAMPs, are passively released from the cell. Studies using *Casp1*^−/−^ macrophages reconstituted with mutant auto-cleaved caspase-1 showed that these cells retained the ability to induce pyroptosis, but demonstrated a lost capacity for IL-1β maturation and release, thereby confirming that caspase-11 can cleave GSDMD independently of the canonical inflammasome and caspase-1 [68].

Caspase-11 activation via engagement with cytosolic LPS has also been reported to induce cleavage of the ATP permeable channel, pannexin-1, leading to release of ATP [69]. The release of ATP subsequently activates the purinergic P2X7 receptor, an essential mediator of K^+^ efflux, which has an essential role in NLRP3 activation, and consequently caspase-1 activation [70,71]. In an experimental model of endotoxic shock, LPS or TLR3-mediated caspase-11 priming, followed by secondary LPS challenge, demonstrate mortality, which is not observed in *Casp-11*^−/−^, *Panx1*^−/−^ or *P2X7*^−/−^ mice. Therefore pannexin-1 and P2X7 appear to be critical downstream molecules in caspase-11 mediated pyroptosis [69]. However, the importance of pannexin-1 during canonical and non-canonical inflammasome activation has been recently challenged by a study that suggests that, while pannexin-1 processing promotes NLRP3 inflammasome assembly during apoptosis, it is dispensable during non-canonical inflammasome activation [72].

Alarmin Release

Alarmins do not contain signaling peptides to induce their cellular release; instead, their release correlates with loss of membrane integrity during pyroptosis. Caspase-11 mediated pyroptosis is therefore required for the release of alarmins, such as HMGB1 and IL-1α [73]. As detailed earlier, HMGB1 has been shown to mediate endotoxin delivery to the cytosolic compartments of macrophage and endothelial cells during sepsis, leading to the activation of caspase-11 and induction of pyroptosis [50]. However, caspase-11 was also observed to mediate HMGB1 release from hepatocytes, driving immune cell pyroptosis in the spleen, peritoneum, and gut [74]. LPS stimulated hepatocyte release of HMGB1, while caspase-11 dependent does not lead to the induction of cell death in hepatocytes themselves, but drives immune cell pyroptosis during sepsis [74]. These studies demonstrate distinct cell-specific roles for HMGB1 in the context of infectious disease.

The processed, 17 kDa form of IL-1α is selectively released from the cell following calcium ionophore stimulation, without any changes in release of the full length 33 kDa form [75]. While calpain has been shown to be responsible for canonical IL-1α cleavage, inflammatory caspases have been implicated in IL-1α secretion from senescent cells that display an altered secretory activity, known as senescence-associated secretory phenotype (SASP) [73]. Murine hepatocytes were shown to require caspase-11 for SASP-driven IL-1α release and immune mediated clearance of senescent cells [76]. Fungal infection studies provide additional evidence for the importance of caspase-11 mediated pyroptosis for alarmin release. BMDM infected with *Paracoccodiodies brasiliensis* were demonstrated to require caspase-11-mediated pyroptosis for the release of IL-1α and the production of nitric oxide, both of which are involved in the restriction of fungal replication [77].

Modulation of Actin Dynamics

During Gram-negative bacterial infection, caspase-11 has been shown to function by promoting phagolysosome fusion, to enable the digestion of phagocytosed bacterium. Amer and colleagues demonstrated that the fusion of *Legionella pneumophila* containing phagosomes with lysosomes is mediated by caspase-11 [78]. Interestingly, caspase-11 was dispensable for the delivery of non-pathogenic bacteria to lysosomes, suggesting that caspase-11 regulates an endocytic route that is distinct from other phagocytic pathways, and specific to pathogenic bacteria. Caspase-11 was shown to mediate phagolysosome fusion and clearance of *L. pneumophila* by promoting actin remodeling and the formation of a filamentous actin (F-actin) network around bacteria-containing vacuoles. Caspase-11-deficient BMDM demonstrated an absence of newly formed F-actin networks, which was accompanied by defective *L.*
*pneumophila* clearance and an accumulation of replicative vacuoles. In support of these findings, phagolysosome fusion has also been shown to be prevented by disorganization of the F-actin network during *Mycobacterium avium* infection [79]. Cofilin is a major actin depolymerization factor that regulates cellular polarity during cellular migration, ruffling of the cell membrane, and driving the leading edge forward via depolymerization of F-actin filaments [80,81]. Dynamic phosphorylation of cofilin regulates actin remodeling in cells. Caspase-11 had previously been linked to the regulation of cofilin-mediated actin depolymerization via its direct interaction with the cofilin activator, actin interacting protein 1 (Aip-1), to promote cell migration during inflammation [81]. The caspase-11 CARD domain was shown to interact with the C-terminal WD40 domain of Aip-1 to influence actin dynamics and cell migration during inflammation, which was independent of the RhoA-Rac-Cdc42 pathway [81]. Amer and colleagues observed that cofilin phosphorylation was impaired in *L.*
*pneumophila* infected caspase-11-deficient BMDM and therefore proposed that caspase-11 functions to activate cofilin-mediated actin remodeling, allowing trafficking of pathogen containing phagosomes to lysosomes, to improve bacterial clearance [78].

A later study from the Amer group suggests that caspase-11 regulates the phosphorylation of cofilin via RhoA GTPase to mediate actin remodeling, phagolysosome fusion, and bacterial clearance during *L. pneumophila* infection [82]. This proposed mechanism is in conflict with the proposal that caspase-11 regulates cofilin activity via its interaction with Aip-1 [81]. Activation of RhoA GTPase, mediated by caspase-11, was shown to promote cofilin phosphorylation and actin depolymerization, whereas caspase-1 was shown to promote cofilin dephosphorylation and actin polymerization. The opposing effects of caspase-1 and -11 are proposed to facilitate the dynamic actin remodeling, which is required for the fusion of *Legionella*-containing vacuoles with lysosomes [82]. Similar findings regarding the importance of caspase-11 for phagolysosome fusion have been observed during infection with the Gram-negative bacteria, *Burkholderia cenocepacia* [83]. Caspase-11-deficient macrophages were shown to exhibit defective autophagosome formation and reduced bacterial clearance of *B. cenocepacia*. The authors propose a role for caspase-11 in autophagy through the modulation of actin dynamics during Gram-negative bacterial infection [83].

In addition to caspase-11-deficient cells having altered migration and reduced fusion of lysosomes to pathogen containing vacuoles [78,81], they have also been shown to have enhanced TCR signaling [84]. This study suggests that caspase-11 negatively regulates CD8^+^ TCR signaling and cytotoxic effector function via reduced T cell expansion/activation, possibly through its ability to regulate actin polymerization [84]. The defective cofilin phosphorylation and actin remodeling observed during caspase-11 deficiency has also been linked to impaired neutrophil migration in an experimental model of gout [85]. In vitro experiments demonstrated that, compared to their WT counterparts, caspase-11 deficient neutrophils had decreased cofilin phosphorylation, impaired migration towards the KC chemokine, and were unable to produce neutrophil extracellular traps (NETs) following stimulation with monosodium urate crystals (MSU) [85].

Altered Mitochondrial Respiration

A recent study has used a model of ischemia (Femoral artery ligation (FAL)) to examine the importance of caspase-1/11 signaling and autophagosome formation in muscle regeneration [86]. Using caspase-1/11 double knockout mice in the FAL model, they showed that these mice had smaller myofibers in regenerating muscles compared to those of similarly treated WT mice. They also observed an altered metabolic profile in muscle satellite cells from caspase-1/11 knockout mice, which exhibited significantly reduced maximal and spare respiratory potential. Non-mitochondrial oxygen consumption and ATP production was similar between WT and caspase-1/11 groups [86]. However, the mechanism underlying the observed dependence for caspase-1/11 signaling in mitochondrial respiration during ischemia is unknown, and further studies to separate the contribution of caspase-1 and -11 to this process are required.

Caspase-11 has also been recently linked with a role in mitochondrial respiration in the context of Gram-positive infection with Methicillin-resistant *Staphylococcus aureus* (MRSA). Mitochondria from caspase-11 deficient macrophages were shown to produce higher levels of superoxide than WT macrophages in response to MRSA [87]. In the absence of caspase-11, recruitment of mitochondria to MRSA-containing vacuoles was shown to be enhanced, which facilitated enhanced mitochondrial-ROS mediated bacteria killing. The actin depolymerization agent, cytochalasin D, was shown to prevent dissociation of mitochondria from MRSA vacuoles, suggesting that the impaired actin dynamics associated with caspase-11 deficiency is responsible for the enhanced association of MRSA vacuoles with mitochondria [87]. Whether this mechanism also led to the enhanced levels of mitochondrial ROS observed in MRSA-infected caspase-11 deficient macrophages or whether caspase-11 has a specific involvement in regulating mitochondrial respiration has yet to be determined.

Activation of Apoptosis

Early studies from the Yuan group showed that, in addition to its importance for pro-inflammatory cytokine secretion, caspase-11 also regulated the activation of caspase-3 and apoptosis in murine models of sepsis and stroke [88,89]. *Casp11*^−/−^ spleen challenged with LPS were shown to have reduced caspase-3 activation. However, spleen cell populations positive for caspase-11 and not capsase-3 were detectable during endotoxin challenge, suggesting that caspase-11 activation does not directly trigger the activation of caspase-3. The authors suggest that inflammatory cell death and apoptosis are linked by a caspase-11 mediated positive feedback loop, rather than initial mechanistic events [88].

An apoptotic role for caspase-11 has also been proposed during endoplasmic reticulum (ER) stress-induced cell death [90,91]. The role of the ER involves protein folding, maturation, storage, and secretion. ER-stress is triggered when its capacity is overwhelmed by cellular demand, and misfolded proteins begin to accumulate. Accumulation of unfolded proteins results in the activation of the unfolded protein response (UPR), which when excessed irreversibly enables cell death. Human caspase-4, which is orthologous to caspase-11, was initially implicated with a role in ER-stress induced apoptosis. A study carried out in 2004 showed that the ER stress inducer, thapsigargin, induced apoptosis in the human neuroblastoma cell line SK-N-SH and that apoptosis was significantly decreased when levels of caspase-4 were reduced using siRNA [91]. The transcription factor C/EBP homology protein (CHOP) can be activated in vitro by treatment with stimuli that induce the UPR and ER stress, such as tunicamycin or thapsigargin. Activation of CHOP by either of these stimuli has been implicated in mediating astrocyte cell death, via activation of caspase-11 and apoptotic caspase-3 [92]. A study using a model of chronic liver disease has similarly reported links between ER stress, caspase-11, and apoptosis. In livers from obese mice, administration of LPS or tunicamycin resulted in UPR activation and CHOP overexpression, which was reported to activate the NLRP3 inflammasome, initiating hepatocyte pyroptosis (via caspase-1 and -11) and apoptosis (via caspase-3 and BH3-only proteins) [90]. However, CHOP activation downstream of ER stress has been very clearly linked to caspase-11 driven pyroptosis, and not apoptosis, in renal epithelial cells, both In vitro and in vivo in models of ischemia-reperfusion and hypoxia reperfusion injury [93]. Owing to the fact that pyroptotic cell death was characterized relatively recently [94], after the publication of many of the papers which report an involvement between caspase-11 and apoptosis, it should be considered that pyroptotic, rather than apoptotic, cell death may have been occurring in some of these studies. As Yuan and colleagues noted in 2002, they observed no evidence of direct activation of caspase-3 by caspase-11, suggesting that apoptosis may occur indirectly, in cells within the vicinity of those that have undergone caspase-11 mediated inflammatory pyroptosis [88].

Caspase-11 Mediated Coagulation

Recent papers have implicated caspase-11 as an initiator of systemic blood clotting and tissue thrombosis. Li and colleagues demonstrated that in vivo activation of the canonical inflammasome (via T3SS Gram-negative rod protein) or the non-canonical inflammasome (via LPS) in mice led to GSDMD cleavage and pyroptotic-dependent release of tissue factor (TF), inducing coagulation and lethality [95]. TF positive microvesicles released into circulation can trigger both arterial and venous thrombosis [96]. In a parallel study, Yang and colleagues highlighted a role for caspase-11 mediated GSDMD cleavage and TF release during disseminated intravascular coagulation (DIC) in sepsis [97]. They showed that GSDMD deficiency inhibits endotoxin induced TF activation by reducing phosphatidylserine (PS) exposure on the outer plasma membrane. Once exposed, PS interacts with TF, resulting in its conformational change and activation. TF acts as a high affinity receptor for factor (F)VII and FVIIa, which serves as the initial activator of the coagulation protease cascade. Caspase-11 mediated GSDMD-N pore formation results in Ca^2+^ influx, which activates transmembrane protein 16F (TMEM16F), a calcium dependent phospholipid scramblase, to mediate phosphatidylserine externalization and subsequent activation of the coagulation cascade [96]. The GSDMD-mediated Ca^2+^ influx also triggers repair programs such as endocytosis of damaged membrane or ectosome membrane shedding, which can negatively regulate pyroptosis downstream of GSDMD [98]. Therefore, while GSDMD-mediated Ca^2+^ influx promotes TF activity, it simultaneously inhibits GSDMD-mediated pyroptosis. This identifies a pyroptosis-independent function for caspase-11, and its substrate GSDMD, during intravascular coagulation.

Levels of IL-1α, IL-1β, and GSDMD activation in septic patients correlate with PS exposure on peripheral leukocytes and DIC scores [96] and exposure of vascular endothelial cells to IL-1α and IL-1β increases TF release [99]. However, deletion of the IL-1R does not prevent LPS mediated coagulopathy in sepsis [96]. Furthermore, caspase-11/GSDMD dependent DIC has been shown to be independent of IL-1 signaling [100]. Therefore, the coagulation process also appears to be independent of IL-1 signaling events downstream of caspase-11.

During *Klebsiella pneumoniae* infection, caspase-11 promotes bacterial clearance from the lungs [101]. *K. pneumonia* infection is commonly associated with pneumonia and sepsis. Caspase-11 deficient mice inoculated with low dose *K. pneumoniae* display increased bacterial colonization in the lungs, corresponding with reduced IL-1α and increased TNF levels in bronchoalveolar lavage fluid and reduced neutrophil infiltration [101,102]. The increased *K. pneumoniae* colonization in caspase-11^−/−^ lungs is also linked to significantly less fibrin formation in the lung, observed by reduced cross-linked fibrin and d-dimer, markers of coagulation [101]. This correlates with findings that thrombin-mediated coagulation is protective against *K. pneumoniae* infection of the lung via fibrin polymerization and platelet–neutrophil interactions [103]. Thus, caspase-11 demonstrates a protective function during *Klebsiella* infection via activation of blood coagulation in the lungs. In summary, these results reveal that caspase-11 and GSDMD are involved in novel mechanisms of coagulation activation and thrombosis as part of the host defense mechanism against bacterial infection.

## 7. Caspase-11 in Disease

As is clear from the functional effects of caspase-11 described above, the major role for caspase-11 is in host protection from endogenous insults and Gram-negative bacterial infections, including *Escherichia coli*, *Citrobacter rodentium,* or *Vibrio cholerae* [73]. Major functions mediated by caspase-11 to facilitate bacterial clearance include cytokine and alarmin release, acidification of bacterial-loaded phagosomes, and pyroptosis, the mechanisms of which are detailed above. When infection becomes systemic, overactivation of caspase-11 pathways leads to pathological inflammation, evidenced by the fact that mice deficient in caspase-11 are protected from models of sepsis, including LPS administration and caecal ligation puncture (CLP) [73,104]. Given the similarities in the features of multi-organ failure that occur during severe cases of SARS-CoV-2 infection and bacterial sepsis, it is likely that caspase-11 and its human orthologs caspases-4 and -5 represent promising targets to limit the excessive inflammatory effects of SARS-CoV-2. It is therefore important that we understand the subtleties that govern their regulation and activation in both healthy and disease scenarios. Although originally limited to bacterial infection, additional disease pathologies have been associated with caspase-11 mediated pyroptosis, including models of lung injury and asthma [38,104], psoriasis in skin [105], and peptic ulcers in gastric tissue [38,106]. In each of these diverse mucosal settings, caspase-11 driven pyroptosis induces the secretion of alarmins that serve to drive and amplify inflammasome activation in immune cells [38,105,106].

Caspase-11 expression levels are stringently regulated and are below detection limits of Northern and Western blot, with transcripts only detectable by reverse transcriptase PCR in most tissues except intestine, where its expression can be detected in unstimulated conditions [88]. Given its protective role against invading bacteria, it is hardly surprising that caspase-11 expression is highest at intestinal mucosa sites. Caspase-11 has been demonstrated to govern pathogen clearance and inflammation in vivo and In vitro in intestinal epithelial cells (IECs), via pyroptosis and extrusion of infected IECs from the intestinal barrier [107]. Our research hypothesized that caspase-11 would therefore be important during intestinal diseases, such as colitis. Using an experimental model of DSS-induced colitis, we and others showed that caspase-11 is also protective in the context of intestinal inflammation [108,109]. Although not related to its protective role during colitis, Demon et al. also show that caspase-11 maintains the integrity of the intestinal microbiome [109]. We have shown that the enhanced susceptibility of caspase-11 deficient mice to DSS-colitis is due to impaired IL-18 production. Impaired IL-18 levels caused reduced intestinal epithelial barrier integrity and decreased cell proliferation, highlighting the importance of caspase-11 in maintaining homeostasis within the intestinal mucosa [108]. Following this line of research, we subsequently investigated the effects of caspase-11 in a murine model of colitis-associated carcinogenesis (CAC) and observed that *Casp-11*^−/−^ intestinal epithelial cells (IECs) were significantly more susceptible to CAC than WT IECs [23]. We observed that colons from CAC-treated *Casp-11*^−/−^ mice had increased expression of proteins associated with early-stage angiogenesis, suggesting that caspase-11 may have a role in inflammation- and cancer-associated angiogenesis. We also demonstrated that the increased susceptibility of caspase-11 deficient mice to CAC was associated with decreased STAT1 activity, identifying a novel anti-tumor role for caspase-11 during CAC via its ability to provide positive enhance STAT1 activation [23]. This evidence points towards a mechanism where caspase-11 controls STAT1 activity through type I IFN signaling, which potentially implicates caspase-11 in the control of an array of anti-tumorigenic and immune effector functions [23]. This murine colorectal cancer model reveals how the amplification of inflammatory signaling via caspase-11 appears to limit tumor initiation and progression, highlighting how context-dependent the inflammatory effects of caspase-11 can be in terms of the cell/tissue type and the stage of disease.

## Figures and Tables

**Figure 1 ijms-22-01506-f001:**
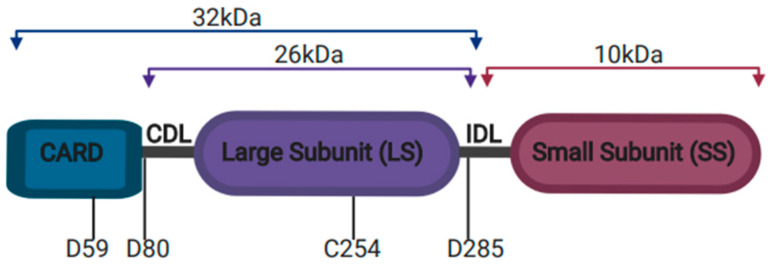
Caspase-11 structural domains. Caspase-11 is comprised of three domains; the N-terminal CARD domain is separated from the large subunit (26 kDa) by a CARD domain linker region (CDL). The large subunit and C-terminal small subunit (10 kDa) are the catalytic domains, separated by the inter-domain linker region (IDL). Caspase-11 auto-proteolysis involves CDL cleavage at D59 and D80 to remove the CARD domain. The large and small catalytic domains are separated by processing of IDL at D285, to generate subunits P10 and P26 or P32. The active cysteine residue C254 is located within the enzymatic active site of the large subunit.

**Figure 2 ijms-22-01506-f002:**
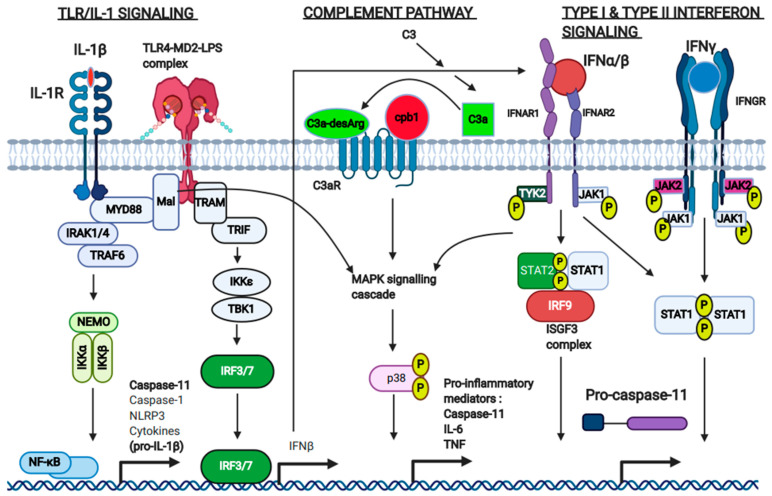
Caspase-11 upregulation occurs as a result of PAMP and/or cytokine signaling and can be amplified by complement signaling. TLR signaling—Exogenous PAMPs, such as LPS, that signal through TLRs result in MYD88-dependant activation of NF-κB. Endocytosed LPS-TLR4 signals through the TRIF adaptor protein, resulting in the activation of IRF3 and IRF7 and activation of type I IFNs, which are responsible for caspase-11 upregulation downstream of LPS. IL-1β signaling—IL-1β signals through the IL-1 receptor complex, which results in MYD88-dependant NF-κB activation and Type I IFN signaling. IL-1β-mediated upregulation of caspase-11 is dependent on type I IFN-signaling. Type I IFN signaling—IFNα/β, which is also generated following LPS and IL-1β stimulation, signals through IFNAR. This results in phosphorylation of STAT1/STAT2 and formation of the ISGF3 transcriptional complex, which significantly upregulates caspase-11. Caspase-11 upregulation via Type I IFNs, LPS, or IL-1β is significantly impaired in *Ifnar*^−/−^ BMDM, highlighting the importance of IFNAR signaling for caspase-11 expression. Type II IFN signaling—IFNγ signals through the IFNGR, resulting in the formation of a phospho-STAT1 dimer, which is capable of directly upregulating genes, such as caspase-11, which have IFN-stimulatory-gene (ISG) promoter sequences. Complement pathway—the complement protein cleavage product, C3a, is modified to C3a-desArg by the carboxypeptdidase Cpb1, which is associated with the C3a receptor (C3aR). The modified cleavage product then signals through C3aR, resulting in downstream activation of the MAPK pathway. This pathway acts to amplify p38 activation downstream of TLR4 and IFNAR signaling, boosting caspase-11 expression.

**Figure 3 ijms-22-01506-f003:**
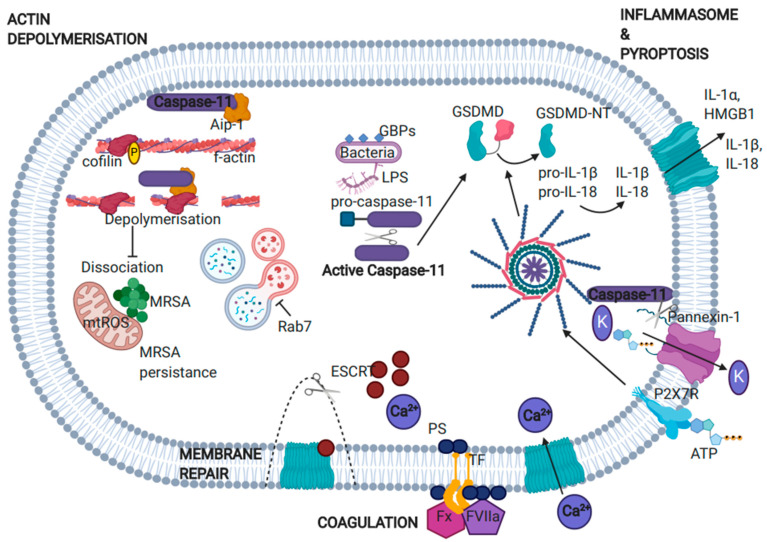
Effector functions of caspase-11. Cytosolic LPS (via GBPs) binds and activates caspase-11, which leads to: Non-canonical inflammasome activation and pyroptosis—direct cleavage of GSDMD by caspase-11, generating the pore-forming P30 N-terminal fragment (GSDMD-NT) and membrane permeable pores, which passively release alarmins, IL-1α, and HMGB1. Caspase-11 activation also triggers NLRP3 inflammasome activation, leading to maturation and release of IL-1β and IL-18. Caspase-11 also cleaves Pannexin-1, mediating ATP release and K^+^ efflux to trigger activation of the NLRP3 inflammasome. Actin polymerization—Activated caspase-11 directly interacts with Aip-1 leading to cofilin dephosphorylation and activation of cellular migration. During Gram-negative bacterial infection, cofilin drives actin remodeling, leading to phagolysosomal fusion and bacterial clearance. However, during MRSA infection, caspase-11 promotion of actin remodeling drives dissociation of mitochondria from MRSA-containing vacuoles, reducing the contribution of mitochondrial ROS to bacterial clearance, enhancing the persistence of MRSA. Coagulation—Caspase-11 and GSDMD-N pore formation mediates Ca^2+^ influx allowing phosphatidylserine (PS) externalization to the outer plasma membrane. Exposed PS interacts with tissue factor (TF), which associates with complement factors to initiate the coagulation cascade. The GSDMD-mediated Ca^2+^ influx also mediates a membrane repair response via ESCRT components, removing GSDMD pores and repairing membrane integrity.

**Table 1 ijms-22-01506-t001:** Stimulators of caspase-11 expression.

Signalling	Receptor	Stimulator	Pathway	Reference
TLR signaling	TLR2	Pam3CSK4	NF-κB activation	[19]
	TLR3	dsRNA	Type I IFN	[19,20]
	TLR4	LPS	Type I IFN	[20,21,22]
	TLR5	Flagellin	NF-κB activation	[19]
	TLR9	CpG-rich DNA	NF-κB activation	[19]
IL-1 signaling	IL-1R	IL-1β	Type I IFN	[23]
Mitochondrial DNA damage	mtDNA	c-GAS-STING	Type I IFN	[24,25]
Complement pathway	C3aR	C3a	MAPK kinase P38 Type I IFN amplification	[26,27,28]
IFN signaling	IFNAR	IFNα/β	Type I IFN	[20,22,23,29,30]
	IFNGR	IFNγ	Type II IFN	[20,23,31,32]

**Table 2 ijms-22-01506-t002:** Regulators of Caspase-11 activity.

Mediator	Description	Effect	Reference
GBPs^chro3^	Facilitate the lysis of OMVs/vacuoles containing LPS/Gram-negative bacteria	Caspase-11 activation significantly impaired in GBP^chro3^ deficient macrophages	[43,44]
IRGMs	Involved in regulating vacuole/lysosome stability	Irgm2^−/−^ BMDMs have heavily exacerbated caspase-11 activation compared to WT BMDMs.	[45,46]
HMGB1	Circulating HMGB1 binds LPS, entering macrophages/endothelial cells via RAGE-mediated endocytosis, promoting lysosomal rupture and LPS release	Resistance to bacterial sepsis in vivo upon neutralization of HMGB1.	[47,48,49,50]
L-adrenaline	L-adrenaline binds ADRA2B, activating PKA. Activated PKA facilitates caspase-11 phosphorylation.	Inhibition of caspase-11 activation. Phosphodiesterase 8A inhibits this effect by degrading cAMP	[51]
Oxidized phospholipids	Oxidized phospholipids interact with caspase-11 by binding to the CARD domain	Low dose; compete with LPS and inhibit caspase-11 activation.	[52,53]
		High dose; induces caspase-11 oligomerization and activation	[54]

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
