# Peer review of "Regulation, Activation and Function of Caspase-11 during Health and Disease"

_ijms, 2021, doi:10.3390/ijms22041506_

Round 1

Reviewer 1 Report

The review by Agnew and coauthors covers several aspects of caspase-11, from the mechanisms regulating its expression and activation to the description of its known and novel functions. The review is very well structured and clearly presented, although unnecessary details may sometimes be distracting (for instance, the mechanism of GSDMD-N insertion into the membrane, lanes 417-424). I have just one comment that the authors may want to address. In the last section (7. Caspase-11 in disease), the authors emphasize the role of caspase-11 in the homeostasis of the intestinal mucosa. I think it would be interesting to mention the interaction with the gut microbiota (for instance, in the same ref 109 cited by the authors).

Others:

- in figure 2, the references included in the legend are sequential with the text (the last ref in the text before Figure 2 is 21, the legend contains the refs 22-24, and the text after the figure starts with ref 25), but this is not the case for figure 3. I would not include references in the figure legends. If they are, please use the same criteria.

- please correct “5” (lane 43). It seems that Ref. 52 is cited for the first time after ref. 92

- the authors should check for typos (lane 96 “capase”, lane 170 and others “E. Coli”, lane 567 “capsase”, lane 598 “vicinty" …) and Greek letters (lane 57 “IFNg”, lane 64 and others “IL-1b”), and define abbreviations at first use (for example “BMDM”).  

Author Response

The authors would like to thank Reviewer 1 for the helpful comments and suggestions. We have addressed each of the points made by Reviewer 1 (R1) below:

Point 1, R1: I have just one comment that the authors may want to address. In the last section (7. Caspase-11 in disease), the authors emphasize the role of caspase-11 in the homeostasis of the intestinal mucosa. I think it would be interesting to mention the interaction with the gut microbiota (for instance, in the same ref 109 cited by the authors).

Response: We have included the findings of ref 109 which identify a role for caspase-11 in regulating intestinal microbial composition (Section 7, line 671-673).

Point 2, R1: - in figure 2, the references included in the legend are sequential with the text (the last ref in the text before Figure 2 is 21, the legend contains the refs 22-24, and the text after the figure starts with ref 25), but this is not the case for figure 3. I would not include references in the figure legends. If they are, please use the same criteria.

Response: We have removed the references from all Figure legends.

Point 3, R1: please correct “5” (lane 43). It seems that Ref. 52 is cited for the first time after ref. 92

Response: The reference 5 (uppercase) has been removed from line 43 and Ref 52 is now in sequence with other refs.

Point 4, R1: the authors should check for typos (lane 96 “capase”, lane 170 and others “E. Coli”, lane 567 “capsase”, lane 598 “vicinty" …) and Greek letters (lane 57 “IFNg”, lane 64 and others “IL-1b”), and define abbreviations at first use (for example “BMDM”).  

Response: These typos have been corrected, many thanks.

Reviewer 2 Report

Re: Manuscript ID: ijms-1097947.

This is a well written review dealing with the complex activities of caspase factors in health and disease, in particular inflammation. The importance of this argument is growing. As a matter of fact, caspase 11 has been the subject of the European project INFLAMMACT, as well. The different arguments are well developed and commented. Minor changes are suggested to improve the paper.

Points of criticism

I suggest the authors to include a short comment on the possible role of caspase in inflammation- and cancer-associated angiogenesis.

A commented table would help the reader to better understand different regulators, stimulators and inhibitors of caspase-11, when listed.

Some reference numbers are not in bold.

Line 132. Replace “Schauvliege” with “Schauvliege”.

In figure 2 “signalling” and “signaling” are both used: choose signalling.

Caption of figure 2, line 146. Replace “IFNa/β” with “IFNα/β”. This must be reported in the figure.

Line 159. Replace “Sander” with “Sander”.

Line 168. Replace “Rathinam” with “Rathinam”.

Line 243. Replace “Shi” with “Shi”.

Line 286. Replace “Finethy” with “Finethy”.

Line 312. Replace “Deng” with “Deng”.

Line 341. Replace “Zanoni” with “Zanoni”.

Line 347. Replace “Chu” with “Chu”.

Line 362. Maybe “cells, where” instead of “cells. Where”.

Line 375. Replace “Chen” with “Chen”.

Line 457. Replace “noncanonical” with “non-canonical”.

Line 533. Replace “gout[86]” with “gout [86]”.

Line 565. Replace “[89], [90]” with “[89,90]”.

Line 603. Replace “noncanonical” with “non-canonical”.

Line 639. Replace “Klebsiella” with “Klebsiella”.

Line 644. Replace “As is” with “As it is”.

Line 656. Replace “hesalthy” with “healthy”.

Author Response

The authors are grateful to Reviewer 2 for their critical evaluation of our manuscript and their helpful suggestions. We have addressed each of the points below:

Reviewer 2 (R2), point 1: I suggest the authors to include a short comment on the possible role of caspase in inflammation- and cancer-associated angiogenesis.

Response: We have included a comment on the potential role of caspase-11 in angiogenesis in section 7, line 680-683.

R2, point 2: A commented table would help the reader to better understand different regulators, stimulators and inhibitors of caspase-11, when listed.

Response: We have included two tables – Table 1 summarises the stimulators of caspase-11 expression and Table 2 summarises the regulators of its activation.

R2, point 3: Some reference numbers are not in bold.

Response: All references are now in bold.

R2, point 4: Line 132. Replace “Schauvliege” with “Schauvliege”, and all other authors names in italics. In figure 2 “signalling” and “signaling” are both used: choose signalling.

Response: All author names that were previously italicised have been corrected. All spellings have been corrected to ‘signalling’.

R2, point 5: Line 356. Maybe “cells, where” instead of “cells. Where”.

Response: We have rephrased these sentences to make the text more coherent.

R2, point 6: Line 450. Replace “noncanonical” with “non-canonical”; Line 522. Replace “gout[86]” with “gout [86]”; Line 555. Replace “[89][90]” with “[89,90]”, Line 592. Replace “noncanonical” with “non-canonical”; Line 628. Replace “Klebsiella” with “Klebsiella”. Line 656. Replace “hesalthy” with “healthy”.

Response: These typos have been corrected in the revised manuscript.